# Neutral and Selective Processes Shape MHC Diversity in Roe Deer in Slovenia

**DOI:** 10.3390/ani12060723

**Published:** 2022-03-13

**Authors:** Elena Buzan, Sandra Potušek, Luka Duniš, Boštjan Pokorny

**Affiliations:** 1Faculty of Mathematics, Natural Sciences, and Information Technologies, University of Primorska, Glagoljaška 8, 6000 Koper, Slovenia; sandra.potusek@famnit.upr.si (S.P.); lukadunis@gmail.com (L.D.); 2Environmental Protection College, Trg Mladosti 7, 3320 Velenje, Slovenia; bostjan.pokorny@vsvo.si; 3Slovenian Forestry Institute, Večna pot 2, 1000 Ljubljana, Slovenia

**Keywords:** major histocompatibility complex, MHC genes, immunogenetics, *Capreolus capreolus*

## Abstract

**Simple Summary:**

Disease prevention and appropriate wildlife management are among the major challenges in wildlife conservation. In the present study, we made a first assessment of the variability of major histocompatibility complex (MHC) genes in roe deer in Slovenia and evaluated local population adaptation by comparing MHC variability with neutral microsatellites. We discovered three new MHC DRB exon 2 alleles in addition to seven previously described in the literature. Moreover, we found evidence of historical positive selection, as selection analysis indicated that approx. 10% of the encoded amino acids were subjected to episodic positive selection. This study provides the basis for further research on immunogenetic variation in roe deer and highlights opportunities to incorporate genetic data into science-based population management.

**Abstract:**

Disease control and containment in free-ranging populations is one of the greatest challenges in wildlife management. Despite the importance of major histocompatibility complex (MHC) genes for immune response, an assessment of the diversity and occurrence of these genes is still rare in European roe deer, the most abundant and widespread large mammal in Europe. Therefore, we examined immunogenetic variation in roe deer in Slovenia to identify species adaptation by comparing the genetic diversity of the MHC genes with the data on neutral microsatellites. We found ten MHC DRB alleles, three of which are novel. Evidence for historical positive selection on the MHC was found using the maximum likelihood codon method. Patterns of MHC allelic distribution were not congruent with neutral population genetic findings. The lack of population genetic differentiation in MHC genes compared to existing structure in neutral markers suggests that MHC polymorphism was influenced primarily by balancing selection and, to a lesser extent, by neutral processes such as genetic drift, with no clear evidence of local adaptation. Selection analyses indicated that approx. 10% of amino acids encoded under episodic positive selection. This study represents one of the first steps towards establishing an immunogenetic map of roe deer populations across Europe, aiming to better support science-based management of this important game species.

## 1. Introduction

Genetic diversity is one of the most important mechanisms that enable a population’s response and adaptation to environmental changes [1,2]. Assessment of genetic diversity (particularly of genes influencing immune response) is therefore urgently needed when developing effective conservation and management programs for wildlife species [3,4]. Since neutral genetic variation provides an incomplete picture of the evolutionary potential of populations (e.g., [5,6]), it is also important to monitor adaptive genetic diversity, which is defined as “genetic variation that produces an advantage in fitness” [7]. Polymorphism in the major histocompatibility complex (MHC), a family of highly variable genes, plays a key role in populations’ resilience against pathogens [8,9,10] as well as in sexual selection, mate choice [11,12,13], fitness, and survival rates [14,15,16,17,18,19]. For this reason, MHC diversity reflects the genetic health of populations [20]. Since the molecular variation among MHC alleles is mainly adaptive and maintained by natural and/or sexual selection [21], these alleles can be considered as non-neutral markers. They have been evaluated in conjunction with neutral genetic markers (e.g., [22,23,24,25,26]) to understand the evolutionary forces (especially the effects of drift vs. selection) acting on populations, and it is this interplay of neutral and nonneutral markers that shapes genetic diversity. On the other hand, neutral genetic markers and genes under selections often provide different insights relevant to predicting adaptive/evolutionary potential; thus, a combined view may be the most informative [2,27] and a multilocus approach is preferable compared with single gene studies [28]. Therefore, when examining MHC diversity, it has been suggested to also focus on other appropriate genetic regions available to expand and complement findings on immune system genetic diversity [29]. Such an approach may be based on a large-scale genomic study of immune genes using quantitative trait loci (QTL), single nucleotide polymorphisms (SNPs), or array-based analyses [30,31]. Comparing results obtained from such different types of markers is important as it has been shown that they may present contrasting results (e.g., [32,33]).

Despite the importance of MHC variability for the evolutionary potential of populations to combat pathogens and the increasing risks posed by (re-)emerging diseases, research on the most variable MHC loci (DRB exon 2) in European roe deer (*Capreolus capreolus*)—the most abundant and widespread large mammal in Europe—is still scarce, with studies currently limited to populations separated by large distances [25,34,35]. For example, Quéméré et al. [25] suggested that genetic drift is the main contemporary evolutionary force shaping immunogenetic variation among three distant (from Chizé, Trois-Fontaines, and Aurignac) roe deer populations in France. Roe deer is a relevant model species for the determination of factors that affect the population genetic makeup: it is widespread throughout Europe [36] and markedly philopatric, i.e., individuals generally maintain small home ranges (usually <100–150 ha) throughout their lifespan (e.g., [37,38]). Moreover, the ecological traits and social behavior of the species are well-known [37,39,40,41,42,43], providing an excellent opportunity to study their relationship with MHC alleles’ diversity.

The objectives of our study were to (i) conduct the first assessment of MHC variation in European roe deer in Slovenia, and (ii) perform this study at a geographical scale (i.e., between the Alps and the Dinaric Mountains) that enables assessment of the effects of selection and gene flow on adaptive variation, i.e., by comparing patterns of MHC allelic and neutral genetic diversity [44].

## 2. Materials and Methods

### 2.1. Ethical Statement

All animals used in the study were legally harvested during regular hunting activities prescribed by the state of Slovenia in annual wildlife management plans. No animal was shot or otherwise killed for the purposes of this study.

### 2.2. Study Area and Sampling

Samples of 156 roe deer female yearlings were collected throughout Slovenia, i.e., in 58 hunting grounds from all 15 Slovene hunting management districts. Therefore, a full range of environmental factors and population traits of roe deer presence in the country was covered (Appendix A).

All samples (either muscle tissue or uteri) were collected by hunters or wildlife researchers immediately after the harvest event within the hunting season (1 September–31 December) in the period 2013–2015, stored in alcohol solution, and frozen until transportation to the genetic laboratory at the University of Primorska.

### 2.3. DNA Extraction and MHC Genotyping by Sequencing

We extracted DNA from tissue samples using a peqGOLD Tissue DNA Kit (S-Line) and followed the manufacturer’s instructions (VWR International, Leuven, Belgium). DNA extraction and quality control procedures are described in detail in Buzan et al. [44].

We amplified a 249-bp fragment of the exon 2 of the roe deer MHC DRB gene using primers LA31 (ATCCTCTCTCTGCAGCACATTTCC) and LA32 (TTCGCGTCACCTCGCCGCTG), which were modified based on primers originally designed for cattle [45]. For the Ion Torrent PGM system, we designed extended primers as follows: forward primer—adapter sequence (30 bp), 10–12 bp barcodes with a specific “GAT” linker to distinguish individuals and primer LA31; reverse primer—adapter sequence and primer LA32 without adapter sequence.

PCR amplification was performed in triplicates in 25 µL reaction mixtures of 50 ng genomic DNA, 0.5 mM of each primer, 0.5 µL dNTPs, 2.5 mM MgCl_2_, 0.5 µL HotStarTaq (Qiagen AllTaq core kit, Qiagen, Germany), 5.0 µL Q-solution, and 6.2 µL H_2_O. The PCR reaction started with an initial denaturation (2 min, 95 °C) followed by 35 cycles of 10 s denaturation at 95 °C, 30 s annealing at 60 °C, and 30 s extension at 72 °C. The final elongation was performed at 72 °C for 10 min. The amplicons from the triplicates were then pooled and purified with magnetic particles Agencourt**^®^** AmPure**^®^** (Agencourt Bioscience Corporation, A Beckman Coulter Company, Beverly, MA, USA). We measured concentrations of pooled and cleaned amplicons by Qubit 3.0 fluorometry using Qubit dsDNA HS (High Sensitivity) Assay Kit reagents (Invitrogen, Carlsbad, CA, USA). Samples were then normalized to 5 ng and combined into a single library. The library was again purified with Agencourt**^®^** AmPure**^®^** magnetic particles. The size and quality of the amplicons were determined using the Agilent DNA High Sensitivity Kit on a 2100 Bioanalyzer (Agilent, Santa Clara, CA, USA), according to the manufacturer’s recommendations. Before loading the library onto the chip, the library was normalized to 100 pM and sequenced using the Ion Torrent S5 on an Ion 530 chip (Thermo Fisher Scientific, Waltham, MA, USA).

For allele calling, we used the pipeline in the Amplicon Sequence Assignment (AmpliSAS) web tool developed for high-throughput genotyping of duplicated polymorphic genes, such as MHC [46]. Filtering of raw data was performed with AmpliCLEAN by removing reads with a Phred quality score <20 and filtering of all reads <250 bp and >300 bp. AmpliSAS clusters true variants with their potential artefacts based on the platform-specific error rates. We used AmpliSAS’s default parameters for Ion Torrent sequencing technology: a substitution error rate of 0.5% and an indel error rate of 1%. An accurate length was required to identify the dominant sequence within a cluster. We did not expect more than two DRB variants per individual, so we kept the “minimum dominant frequency” clustering threshold at 25%, based on previously published work on roe deer [25,34,35,47]. We discarded variants with a frequency <1% within an amplicon. True variants of the DRB exon 2 fragments were aligned and translated into protein sequences to check for evidence of pseudogenes, such as the presence of premature stop codons or indels. A maximum of 5000 reads per amplicon was used for genotyping to reduce computational load, and we repeated the analysis three times.

### 2.4. MHC Diversity

#### 2.4.1. Sequence Diversity

The unique sequences were aligned, edited, and confirmed to be roe deer MHC DRB exon 2 alleles using MEGA X [48] by comparing them with alleles downloaded from GenBank (Appendix A). DnaSP v.6.12 [49] was used to calculate the average number of nucleotide differences (k) and number of segregating (variable) sites (S). The average pairwise nucleotide distances (Kimura 2-parameter model; K2P) and Poisson-corrected amino acid distances were calculated in MEGA for the following: overall, Antigen Binding Sites (ABS), and non-ABS. The locations of the putative ABS and non-ABS were inferred from the human MHC II molecule structure [50].

#### 2.4.2. Clusters/Populations Genetic Parameters and Molecular Variance

Values for nucleotide diversity (π), 4Nμ for autosomal genes of diploid organism theta (θ), Tajima’s D based on the site frequency spectrum from DNA sequences, and allele number (A) were calculated using DNA SP. We visualized spatial patterns of genetic diversity across the landscape by interpolating and mapping the allelic richness (AR) for each population using the SPADs and R version 4.0.5 [51]. Observed and expected heterozygosity values (Ho and He) and pairwise F_ST_ between groups were calculated and tested by Arlequin version 3.5 [52].

We tested conformance of the allele frequencies with Hardy–Weinberg expectations, both within each of the three clusters (defined by neutral markers; Appendix A; [44]), with groups/populations defined by geographical feathers, and overall, by using the complete enumeration algorithm of Louis and Dempster [53] as implemented in Genepop 3.5 [54]. In Genepop 3.5, tests addressing specific hypotheses of heterozygote excess and deficit were performed, and differences in allele and genotype frequencies between clusters/groups were assessed with Markov chain Monte Carlo approximations of Fisher exact tests [54].

MHC-based population structure in the study area was assessed by analysis of molecular variance (AMOVA; [55]) implemented in Arlequin with 10,000 permutations and by using program STRUCTURE 2.3.4 [56]. Mantel tests were performed in adegenet in R to examine the isolation-by-distance relationship between estimates of MHC F_ST_/(1 − F_ST_) and the natural logarithm of the geographic distance.

We found no evidence for population genetic differentiation at MHC DRB exon 2 alleles (Appendix A); for further analysis, we pooled individuals based on the location of their death into (i) ten predefined groups based on the different historical management of roe deer and geographical characteristics of Slovenia (for the purposes of the study, also called populations) and (ii) three genetically differentiated clusters revealed by neutral genetic markers (for details, see Appendix A, Appendix A, and Buzan et al. [44]). This partitioning is important for science-based population management within the country as it provides additional insight into the adaptive immunity of roe deer within hunting management units.

#### 2.4.3. Clustering of DRB Exon 2 Alleles

The evolutionary relationships between all DRB exon 2 alleles were analyzed by a median-joining network implemented with the software Network 4.6.0.0 (http://www.fluxus-engineering.com/sharenet.htm; accessed on 20 September 2021). All positions were equally weighted and the ε parameter was set to zero.

We also included published roe deer DRB exon 2 allele sequences from GeneBank, which were not found in Slovenia: Caca-DRB*0101, Caca-DRB*0202, Caca-DRB*0203, Caca-DRB*0204 (Appendix A; [25,34]); in the analysis, they were represented as one individual. The MJ network was rooted using haplotypes of red deer (*Cervus elaphus*; GenBank number EU573258). We attempted to study functional MHC DRB diversity by clustering alleles into supertypes (i.e., groups of alleles assumed to have similar peptide-binding capacities). Therefore, we performed clustering based on amino acid polymorphism at the positively selected amino acid sites (PSSs). For the PSSs, we retained all codons with Bayes Empirical Bayes (BEB) posterior probability >95% in the M8 model (Appendix A). The PSS of each allele was numerically characterized by a set of five physicochemical descriptors for each amino acid. To cluster the alleles into supertypes, we performed DAPC using the “adegenet” package. First, we defined clusters (k) using the find.clusters() function and kept all principal components (PCs). We compared different clustering solutions based on the Bayesian Information Criterion (BIC) values for increasing number of clusters. According to recommendation, the optimal number of clusters must be chosen as the number of clusters with the lowest BIC value, after which the BIC value decreases by a negligible amount. However, as the BIC values steadily decline, there was no point of stabilization or ‘elbow’ at which the value began to increase (Appendix A). The likely reason for such result is that only six sites showed signs of positive selection, which did not provide enough resolution for reliable supertyping. We therefore decided that downstream analyses based on supertypes were not relevant and used a simplified approach that included alleles.

#### 2.4.4. Detecting Signatures of Recombination and Selection in MHC DRB Exon 2 Alleles in Roe Deer

The codon-based Z-test of selection (implemented in MEGA X) was used to calculate the average nonsynonymous/synonymous substitution rate ratio (dN/dS = ω) across the entire sequence using the counting method of Nei and Gojobori [57], corrected for multiple substitutions [58].

Using a one-tailed Z-test with standard errors resulting from 10,000 bootstrap replicates, we calculated the rate of positive selection sites separately over the entire DRB exon 2 sequences and over the extracted antigen binding codons. We also used an approach with a more powerful maximum-likelihood method that allows the dN/dS ratio to vary among codon sites; this was implemented in the EasyCodeML program [59], which identifies codons affected by positive selection based on a Bayesian approach and was used in additional analysis for testing historical selection. The models implemented in this study were M2a and M8. Furthermore, we assessed the influence of positive selection on individual codons using single-likelihood ancestor counting (SLAC), fixed effects likelihood (FEL), mixed effects model of evolution (MEME), and fast unconstrained Bayesian approximation (FUBAR) methods performed in the Datamonkey 2.0 server (http://www.datamonkey.org/; accessed on 20 September 2021; see also [60]).

Several methods were implemented to detect the presence of recombinant sequences, which may generate false positives when determining positive selection in our dataset. Namely, we used the online program GARD (genetic algorithm recombination detection [61]) at the Datamonkey website and the Recombination Detection Program, version 4 [62] program suite, which implements a number of methods (RDP [63], BOOTSCAN [64,65], GENECONV [66], MAXCHI [67,68], CHIMAERA [67], SISCAN [69], and 3SEQ [70].

### 2.5. Determination of Genetic Variability among Groups and Clusters Based on Neutral Microsatellite Loci

We reran the analysis on 156 genotyped yearlings, previously used in Buzan et al. [44]. Population structure was determined using program STRUCTURE 2.3.4 [56,71]. Mean number of alleles (A), allelic richness (AR), observed (Ho) and expected (He) heterozygosity, deviation from Hardy–Weinberg equilibrium, pairwise F_ST_, AMOVA, and isolation-by-distance were estimated using the same software and protocols described by Buzan et al. [44].

## 3. Results

### 3.1. Diversity of MHC DRB Alleles

We found 10 functional alleles for MHC DRB exon 2 coding for different amino acid sequences among tested individuals. No evidence of multiple locus amplification was found, confirming previous reports for roe deer [25,34]. A summary of the allele variants found in each individual is provided in Appendix A. The three novel DRB exon 2 sequences identified in this study were named in accordance with previously established nomenclature as Caca-DRB*0402, Caca-DRB*, and Caca-DRB*0404, and were deposited in GenBank with accession numbers from OL355104 to OL355106. One of the new alleles, Caca-DRB*0402, has a deletion in codon 65 (del65), which has already been described in alleles Caca-DRB*0202, Caca-DRB*0301, Caca-DRB*0302, and Caca-DRB*0303 [25,34,35].

The sequence of the remaining seven alleles found in roe deer in Slovenia (Caca-DRB*0102, Caca-DRB*0201, Caca-DRB*0301, Caca-DRB*0302, Caca-DRB*0303, Caca-DRB*0304, Caca-DRB*0401) matched 100% with alleles in previously published studies [25,34] (Appendix A).

Out of the 156 genotyped roe deer, 65 (46%) were homozygous, with 38 (31%) of them being homozygous for the most common allele Caca-DRB*301. Interestingly, no homozygous individual with the alleles Caca-DRB*0401, Caca-DRB*0303, or Caca-DRB*0402 was found (Appendix A). The two most common alleles in all groups/populations were Caca-DRB*0301 (with the highest frequency: 39%) and Caca-DRB*0302 (27%). The eight other alleles had a frequency lower than 8% (Appendix A), with six alleles being relatively common in all populations and the other two (Caca-DRB*0201; Caca-DRB*0303) being found in only a few populations (Appendix A). Caca-DRB*0303 was present only in S1 and C5 populations and Caca-DRB*0403 was not found in C1, N2, and N3 populations. The newly discovered allele Caca-DRB*0402 was present in all populations, except S1.

An analysis of nucleotide alignment revealed 22 segregating (variable) nucleotide sites distributed across 83 codons. The overall nucleotide evolutionary distance and the amino acid evolutionary distance were 5% and 9%, respectively. The average number of nucleotide differences among alleles was k = 9.833 (Table 1).

### 3.2. Clusters/Populations Genetic Parameters and Molecular Variance

We found no evidence for population genetic differentiation at MHC DRB exon 2 alleles (Appendix A). Table 2 lists the values of diversity parameters for three genetic clusters of roe deer in Slovenia as recognized by neutral microsatellite diversity data (Appendix A, Appendix A, [44]). The numbers of MHC alleles ranged from 8 in the southwestern cluster to 10 in the central cluster. Allelic richness was similar in all clusters (8.000–8.602). The expected heterozygosity was lower in the southwestern (0.414) than in other two clusters (0.524–0.593), but θ was higher in the first one (0.022) than in the central and the north-eastern clusters (0.017). Tajima’s D values were positive in all clusters and significant for overall data. The average pairwise F_ST_ values between southwestern and central clusters were negative (equal to zero) and nonsignificant (−0.0029), while between southwestern/northeastern and central/northeastern clusters were positive but also nonsignificant (0.0089 and 0.0002, respectively). The average nucleotide diversity was π = 0.045, and was very similar in all three clusters (Table 2).

For the ten groups/populations (Appendix A), the numbers of alleles in the populations ranged from 4 in C1 to 10 in C4 (Figure 1, Appendix A). Allelic richness was the highest in C2 (AR = 5.480) and the lowest in C1 (4.000). Observed heterozygosity in one of the northeastern (N2) population and the southwestern (S1) population were lower (He = 0.471 and 0.414, respectively) compared with other populations (0.672–0.849). Nucleotide diversity was the highest in C3 (π = 0.050) and the lowest in N3 population (0.039). Parameter θ varied from 0.022 to 0.030: it was lowest in C4 and C3 and highest in C1 and C2 populations. Tajima’s D values were positive in most populations except C1 and N3, which indicates the possibility of a recent population increase after a bottleneck. The average pairwise F_ST_ values between most of the populations were negative and nonsignificant (Appendix A), which means there is no genetic subdivision, except between populations N4 and C2, where the F_ST_ value was positive and significantly different.

Hierarchical AMOVA revealed that most of the genetic variation in the population structure can be explained by within-population differences (98.9%), while only 0.7% of MHC variability is due to differences among populations within three clusters. However, this estimate was not statistically significant (Appendix A). We did not detect significant isolation-by-distance among populations (Mantel test: r = 0.211, *p* = 0.406).

### 3.3. Recombination and Natural Selection in MHC DRB Alleles in Roe Deer

In recognized alleles, the overall number of codons was 83, of which 67 (81%) were non-ABS amino acids and 16 (19%) were ABS. Negative ω (dN/dS) for non-ABS indicates possible selective removal of deleterious alleles, which ultimately reduces variation in a population. The global estimates of ω, averaged across all codon sites using the codon-based Z-test of selection, showed the presence of positive selection at the DRB locus. The nonsynonymous mutation rate (dN = 0.05) exceeded the synonymous one (dS = 0.02) (Table 3). Methods calculating ω values on individual codons (models M2a and M8) identified up to eight codons predicted to be affected by positive selection. The selection models revealed different levels of selection pressure at the loci analyzed. The DRB locus showed signs of strong selection pressure with six positively (posterior probability >99%) selected codon sites (Table 4). Mean values of ω for individual codons at each locus are presented in Appendix A. Recombination analysis revealed two recombination events that resulted in two alleles originating from their combination process (Table 5).

### 3.4. Evolutionary Relationships among DRB Exon 2 Alleles in Roe Deer

The median-joining (MJ) network among 14 alleles (eleven previously described [25,34,35] and three novel ones) showed that the number of mutational steps between adjacent alleles ranged from one (several cases) to eight (between Caca-DRB*0401 and Caca-DRB*0403) (Figure 2). The maximum number of mutational steps across the network between Caca-DRB*0101 and Caca-DRB*0404 was 42, with one hypothetical allele between Caca-DRB*0202 and Caca-DRB*0203/Caca-DRB*0304. The MJ network grouped seven alleles with a deletion of codon 65 close together in one group and the other five without deletion of this codon in a separate group.

### 3.5. Genetic Diversity and Genetic Structure Based on Neutral Loci

Reanalysis of genotyped data on 156 yearling females confirmed previously published results on 241 roe deer individuals, including adults [44]. The best population assignment model resulting from the structure analysis (using K after [56]) divided individuals into three groups (K = 3), which matched fairly well with their geographical origin (Appendix A, Appendix A) as well as with the population structure previously found using a more heterogenous sample set [44].

The average number of alleles per population ranged from 3.3 to 6.0. Due to smaller sample size, allelic richness across populations (3.58–3.88) was lowest as found previously [44], with the highest value in Slovenske gorice/Prekmurje (N4) and the lowest in the coastal region (S1). A similar pattern was also found for Ho (0.58–0.67) and He (0.60–0.66), with Dinaric Mountains population (C3) having the lowest Ho value (Appendix A). The three genetic clusters defined by STRUCTURE had very similar genetic diversity parameters (Ho varied between 0.61 and 0.66 and AR between 6.02 and 6.12, respectively; Appendix A). Pairwise F_ST_ values between groups/populations ranged from 0.004 to 0.069, and more than half of them were significantly different from zero (52%; Appendix A). The highest F_ST_ value was observed between population S1 (coastal Slovenia) and one of the central (C2) and northern (N4) populations, i.e., similar to those found by Buzan et al. [44]. Between genetic clusters, the F_ST_ values were not significant (Appendix A).

On the other hand, AMOVA did not strongly support the three-group structuring revealed by the STRUCTURE, as the among-group variance was low and not significant. Nevertheless, it supported a significant association of individual’s genotype with its geographical position, as differences among populations within groups were significant (Appendix A; [44]).

Microsatellite genetic distances between individuals were positively correlated with the geographical distances between them (Mantel test: *t* = 10.03, *p* < 0.001, R^2^ = 0.008). Although this finding confirms to some extent that the geographical distance among individuals has some effect on the spatial genetic structure of roe deer in Slovenia, its influence is rather weak, which was already shown by Buzan et al. [44].

## 4. Discussion

Our study confirmed that roe deer in the study area (i.e., the contact zone between the Alps and the Dinaric Mountains in central Europe, where data on the MHC gene polymorphism of this species have been completely lacking) have maintained a high level of MHC diversity compared with those in northern Europe [34] despite reductions in abundance and effective population size in the past [73]. This may be due to strong historical demographic fluctuations consistently attributed to founder events [74] and has also been previously described in roe deer from France [25,75]. This coincides with a high variation in biochemical markers in roe deer from central Europe [76], which Mikko et al. [34] already hypothesized should lead to more MHC alleles being found there.

Roe deer in Slovenia as well as other roe deer populations studied [25,34,75] have lower MHC diversity compared with other wild ungulates (Alpine chamois (*Rupicapra rupicapra*) [77]; Pyrenean chamois (*Rupicapra pyrenaica*) [78]; red deer [26,79]; wild boar (*Sus scrofa*) [80]). This is probably an influence of Pleistocene glaciation [34,81], when gene pool depletion was caused by inbreeding combined with genetic drift [82]. However, the history of the species in the study area probably also plays an important role, since, during the 20th century, roe deer from the central part of Slovenia increased its range both in the Sub-Mediterranean and Karst regions in the south, as well in the open agricultural areas of the Sub-Pannonian region in the northeast [44]. Therefore, gene flow has been primarily limited within our study area (all of Slovenia), which is also supported by the pronouncedly philopatric spatial behavior of the species (e.g., [37,38]).

The most common allele found was Caca-DRB*0301 with a deletion of codon 65, which was rare in northern Europe [34]. Interestingly, four other alleles with a deletion of codon 65 were found in Slovenian roe deer: three have already been found in France [25], but Caca-DRB*0404 is a new one. Mikko et al. [34] suggested that these alleles evolved from a common ancestor of the Bovidae and Cervidae (approx. 20 million years ago). The presence of del65 alleles in these two clearly distinct families suggests the possibility of a cross-species mode of inheritance of a del65 allelic lineage that arose before the split of bovids and cervids [35]. However, it is extremely difficult to determine with certainty whether shared sequence motifs between unrelated MHC alleles are due to common ancestry or convergent evolution [83]. Nevertheless, the del65 mutation is interesting from a functional point of view because it could affect antigen binding of exon 2 of the DRB gene [35,84]; thus, the mutation appears to be functionally acceptable or even favored.

The evolutionary distances between nucleotides and amino acids were much higher in ABS than in non-ABS (17% vs. 2%; 43% vs. 3%), indicating a direct effect on binding properties for antigenic peptides [85]. Mutations in the ABS domain are usually associated with differential disease resistance [86]. The observed DRB exon 2 alleles in roe deer showed similar nucleotide and amino acid divergence at the ABS and non-ABS positions as previously described by Mikko et al. [34] and Quéméré et al. [25].

### 4.1. Genetic Diversity and Population Structure

The lack of spatial genetic differentiation in the MHC DRB is interpreted as a consequence of balancing selection where genetic structuring at the MHC loci is expected to be low because MHC polymorphism is maintained over the long-term across populations, even in the case of restricted gene flow [23,87]. In our study, this is supported by STRUCTURE and AMOVA analyses (Appendix A, Appendix A), a positive and significant Tajima’s D, and lower nucleotide diversity compared with haplotype diversity (Appendix A).

Probably, MHC genetic diversity was affected by balancing selection and to a lesser extent by neutral processes, but we did not found evidence of local adaptation, which has already been suggested for roe deer by Quéméré et al. [25]. Our conclusion is supported by several facts that showed weaker patterns of genetic structuring for MHC compared with microsatellite loci (for the latter, see also [44]): (i) Cluster analyses revealed no structure in the MHC (Appendix A, Appendix A), whereas three evident genetic clusters were found using microsatellites (Appendix A). (ii) Only one of the pairwise F_ST_ comparisons was significant for MHC (between C2 and N4 populations), while F_ST_ values were much higher for microsatellites and were significant in 22 cases, i.e., 52% (Appendix A, Appendix A). (iii) We found no evidence of isolation by geographic distance for MHC (*p* = 0.41), while a strong and significant pattern was confirmed for neutral microsatellite loci (*p* < 0.001; see also [44]).

When roe deer individuals were grouped into three clusters defined by microsatellite data (Appendix A), the number of MHC DRB alleles and observed heterozygosity did not differ significantly among those clusters (Table 2). In contrast, the number of MHC DRB haplotypes varied significantly among ten groups/populations predefined on the basis of differences in historical management and geographic characteristics of Slovenia (Appendix A). In all studied populations, we found moderate MHC diversity (He = 0.286–0.833) relative to the observed heterozygosity in other European roe deer populations, which ranged from low in northern Europe [34] to relatively high in France [25]. We also found the same number of alleles (10) and polymorphic sites (22), but two more nonsynonymous sites, as in the three populations from France [25], whereas only four alleles were found in northern European populations, i.e., in Norway and Sweden [34].

Heterozygosity and allelic richness were maintained in all populations (Table 2), despite demographic changes, which were proven with the neutral genetic diversity data (Appendix A and [44]). Isolation and significant effects of genetic drift may contribute to lower allelic richness in the Dinaric Mountains. This may be due to environmental conditions unsuitable for roe deer in this region, i.e., the predominance of dense old forests with a closed canopy (mainly Abieti-Fagetum), which cover tens of thousands of hectares along this mountain chain [72]. The low allelic richness in one of the southern populations (S1) and the sub-Pannonian population (N2) (Figure 1) may also be related to the history of the species in the 19th century, when roe deer in southern Slovenia experienced a genetic bottleneck, while the sub-Pannonian populations had a smaller effective size and were less likely to require individuals to disperse over long distances and/or roam over agricultural land, which is also indicated by neutral genetic diversity data.

As most wildlife populations have undergone complex historical demographic changes and range shifts in recent decades in response to human-mediated land-use changes, it is necessary to clarify their consequences on adaptive genetic diversity [88]. Due to past bottlenecks and recent founder effects, large shifts in allele frequencies can occur on the colonization front without selection. The low MHC diversity observed in the most southern population could also be due to declining parasite-mediated selection [89]. Lower parasite pressure during the colonization process in the dry and warm sub-Mediterranean climate may have caused roe deer to shift energetic resources from host immunity to reproduction [90]; however, we believe this was not the case in Slovenia, as the reproductive potential of the species in the coastal area is lower than in other parts of the country, particularly in the northeast [44,91]. Thus, genetic drift and migrations could be the dominant contemporary forces shaping MHC variation in roe deer (Table 4). Positive selection most likely affects only a few sites over time [92,93]. We detected sites under positive selection and an excess of nonsynonymous substitutions in amino acid residues, suggesting that balancing selection played a role in maintaining this MHC sequence polymorphism. This result is consistent with most of the empirical and theoretical work on MHC in small populations or those experiencing bottlenecks [94,95].

### 4.2. Selection and Recombination

Both maximum-likelihood codon-based selection models support historical selection at the MHC DRB exon 2. Importantly, all codons detected under positive selection were at ABS sites [96], which are involved in pathogen binding recognition [97]. Recombination was detected at one site near an ABS codon. We found a signature of positive selection by the overall ratio ω = 1.40, which is similar to that found in other wild ruminants in Europe (Alpine and Pyrenean chamois [78]; red deer [26,79]). Although our results indicate that positive selection likely played an important role (the amino acid in position 86 was proved to be under positive selection by four of our tests; Appendix A), this does not mean that selection is still acting on current populations. The excess of nonsynonymous mutations may take many generations/cohorts to disappear after the selection process stopped [98]. Short-lived balanced polymorphisms favor rapid adaptive response of populations to changing selection pressures through small adjustments in the level of gene expression [99] and occur in high frequency in the genome, especially in alternative combinations of alleles involved in pathogen resistance [100].

The multigene/multipathogen association approach developed by Quéméré et al. [75] suggested that, in roe deer, innate immunity plays a key role in pathogen-mediated directional selection rather than heterozygote advantage in MHC genes. However, they also stressed the limitation of their work because of insufficient statistical power to test the role of rare variants and the lack of a long time series [101]. In line with this, our study also revealed a strong need for continuing research into MHC genes, their diversity, and consequences for the resilience and fitness of individuals. Indeed, by employing the most important game species (e.g., roe deer or other wild ungulates) as models it may be possible to overcome the issue of a large sample set covering longer time series.

## 5. Conclusions

Our results suggest that MHC DRB exon 2 diversity in roe deer was influenced by balancing selection and to a lesser extent by neutral processes; however, we did not find evidence of local adaptation. Overall, this study contributes to a greater understanding of how roe deer (populations) respond to selection pressures across their range. However, further research would be needed to investigate the relationships between immune genetic diversity and pathogen resistance, particularly with respect to habitat variation in biotic and abiotic factors that likely support different pathogen communities and, thus, different pathogen-mediated selection systems. Analysis of MHC variation provides a good framework for studying the local adaptations and genetic health of wildlife populations but provides only a partial understanding of how species adapt to exposure to diseases and/or other stressors [75]. Therefore, studies of other genes associated with innate immunity are needed to provide a more comprehensive understanding of the genetic adaptive potential of species.

Further studies are also needed to investigate the association between the genetic diversity found here and parasite tolerance/resistance; body condition; and ultimately, individual fitness, with the aim of elucidating the mechanisms of balancing selection such as heterozygote advantage and spatiotemporal fluctuating selection [102]. Such research may have important implications for wildlife management and epidemiology [29] but may also help to reduce risks for human and domestic animal health from a One Health perspective.

## Figures and Tables

**Figure 1 animals-12-00723-f001:**
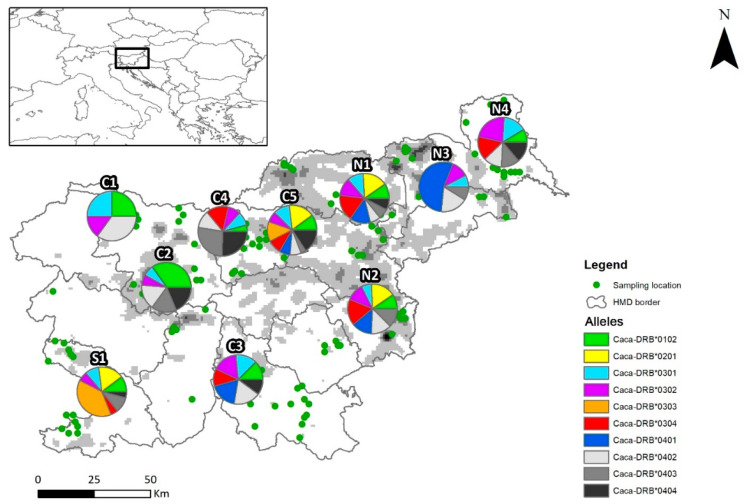
Sampling locations (green dots) and DRB exon 2 allele frequencies of roe deer females in the study area (whole Slovenia). Pie diagrams show DRB allele frequencies of each studied group/population defined by geographical features of Slovenia (see Appendix A for details of the studied individuals and names of the populations). Grey background on the map indicates the gradient of roe deer population density (white: 0–9 animals/km^2^; black: 40–49 animals/km^2^; sensu [72]), and lines separate 15 hunting management districts.

**Figure 2 animals-12-00723-f002:**
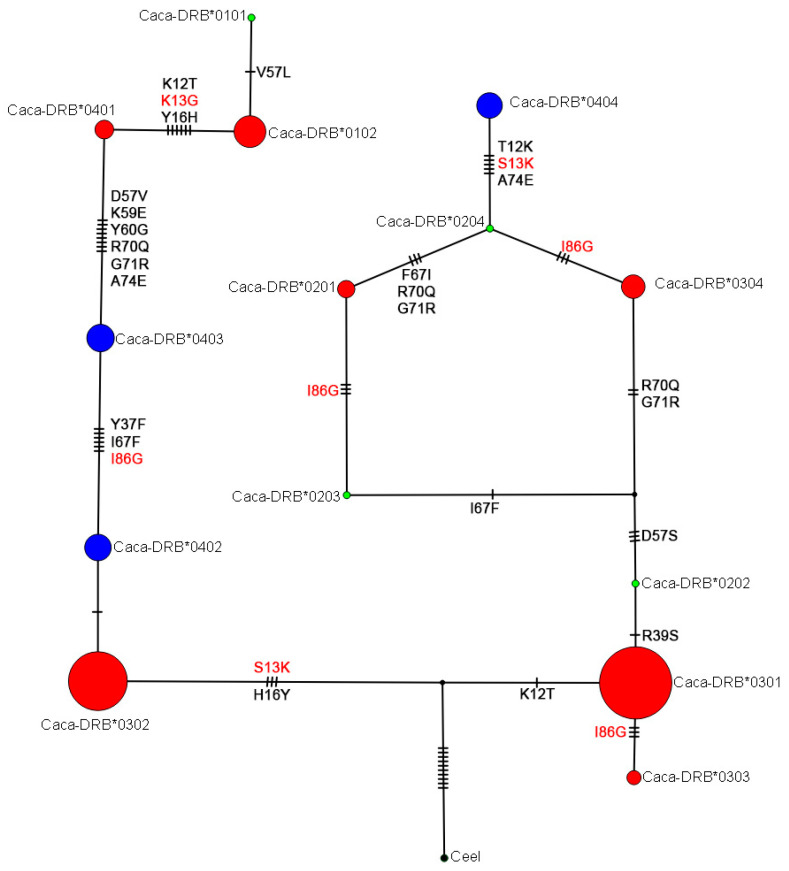
Evolutionary relationships among DRB exon 2 alleles in European roe deer represented by a median-joining network. Alleles are indicated by circles whose size is proportional to the number of roe deer individuals. Number of mutations separating nodes is represented by slashes that cross network branches. A small black circle indicates a hypothetical allele. Alleles marked in blue have been found only in Slovenia (our study), red-marked alleles are known in roe deer from other countries/studies and were also found in Slovenia, and green-marked alleles were not found in our study (represented in the network only by one individual per allele). Nonsynonymous changes are red-marked and indicated by amino acid change (protein variants) coded from A to Q [50]. Amino acid position 86 (nucleotides position 256/257) was proved to be under positive selection by four of our tests.

**Table 1 animals-12-00723-t001:** Nucleotide and amino acid diversity of all MHC DRB alleles found so far in roe deer in Europe (detected in this study and downloaded from GenBank). Kimura 2-parameter model with a gamma distribution shape parameter (K2P) was used to calculate overall nucleotide evolutionary distance and the Poisson substitution model was used for calculating the amino acid evolutionary distance.

Parameters Related to Nucleotide Differences		Overall Nucleotide Evolutionary Distance			Amino Acid Evolutionary Distance		
k	S	All Sites	ABS	Non-ABS	All Sites	ABS	Non-ABS
9.833	22 + 1 indel (13 + 1 indel) *	0.05 (0.01)	0.17 (0.06)	0.02 (0.01)	0.09 (0.03)	0.43 (0.22)	0.03 (0.02)

Notes: k—average number of nucleotide differences; S—the number of segregating sites; ABS—antigen binding sites; non-ABS—nonantigen binding sites. * SD is given in parenthesis, except for S, for which the number of nonsynonymous sites is provided.

**Table 2 animals-12-00723-t002:** MHC DRB exon 2 genetic diversity of roe deer in Slovenia (overall and within three K-clusters).

Geographical Group	Abbr.	n	A	AR	Ho	He	π (SD)	θ	Tajima’s D
Overall		156	10	10.000	0.529	0.739	0.045 (0.004)	0.015	2.263
Southwestern	SW	29	8	8.000	0.414	0.713	0.028 (0.004)	0.022	0.579
Central	C	68	10	8.357	0.524	0.724	0.029 (0.002)	0.017	1.610
Northeastern	NE	59	9	8.602	0.593	0.759	0.027 (0.002)	0.017	1.428

Notes: n—number of individuals; A—number of alleles; AR—allelic richness; Ho—observed heterozygosity; He—expected heterozygosity; π—nucleotide diversity; θ—4 Nμ for autosomal genes of diploid organisms; Tajima’s D (value in bold is significant; *p* < 0.05). Deviation from Hardy–Weinberg equilibrium was significant for overall and cluster data.

**Table 3 animals-12-00723-t003:** Relative rates of nonsynonymous (dN) and synonymous (dS) substitutions (with standard errors) calculated in roe deer DRB exon 2 alleles (found in this study and previously published in [25,34]) for antigen binding sites (ABS) and non-ABS. Statistical significance (*p*-value) was tested using the one-tailed Z-test with standard errors resulting from 10,000 bootstrap replicates.

Main Parameters	Overall	ABS	Non-ABS
**N**	83	16	67
**dS (SE)**	0.02 (0.01)	0.02 (0.02)	0.03 (0.01)
**dN (SE)**	0.05 (0.02)	0.20 (0.08)	0.02 (0.01)
**ω dN/dS)**	1.40	**2.60**	−0.40
** *p* ** **-value**	0.08	0.01	1.00

Note: N—number of codons. Significant value is bolded.

**Table 4 animals-12-00723-t004:** Codon sites under positive selection as predicted by codon evolution models M2a and M8 using the Empirical Bayes approach in EasyCodeML.

Codon Sites under Positive Selection	Selection Model
12, **13**, **16**, **57**, 67, 70, 71, **86**	M2a
12, **13**, **16**, **57**, **67**, **70**, 71, **86**	M8

Note: The codon sites (del65 is included in the numbering) inferred to be under selection with posterior probability >99% are listed in bold, and sites with posterior probability >95% are in standard font. ω values that represent synonymous vs. nonsynonymous substitutions (dN/dS) for particular codons are given in Appendix A.

**Table 5 animals-12-00723-t005:** Recombination events.

Event	Recombinant Sequence	Major/Minor Parent	Consensus Score	Beginning/Ending Breakpoint	Probability (MC Corrected) *	Methods
1	Caca-DRB*0403	Unknown/ Caca-DRB*0401	0.534	170/242	0.0235	MaxChi, SiScan, 3Seq
2	Caca-DRB*0401	Caca-DRB*0404/Unknown	0.481	84/266	0.0177	MaxChi

* Corrected for multiple comparison.

## Data Availability

All attributive data are given in Appendix A.

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
