# Peer review of "Neutral and Selective Processes Shape MHC Diversity in Roe Deer in Slovenia"

_animals, 2022, doi:10.3390/ani12060723_

Round 1
Reviewer 1 Report
The manuscript presented by Buzan et al. describes the diversity of MHC class II, DRB locus in roe deer in Slovenia. The assessment of MHC diversity, especially DRB locus for wild mammal populations, has been used for the last two decades as an important factor in determining population condition and its future prospects. As a consequence, the diversity of this region is characterized by a high number of populations and species. I agree with the authors of the manuscript that adding additional sequences to this database is valuable information, but I can’t consider the manuscript as innovative and of wide interest to the scientific community. I also found some major issues in data analysis and interpretation of the results.
Below I present some specific comments to the authors:
- When the manuscript is published I would change a title adding also the information on forces shaping the diversity of the region, as this was also the aim of the manuscript.
- Line 28; the lack of population genetic differentiation in MHC is not an evidence of balancing selection itself. It is when different from the structure displayed by neutral markers.
- In the introduction, the authors underline the importance of MHC diversity for assessing the evolutionary potential of populations and its usefulness for many types of evolutionary studies. This is all true, but they don’t mention the number of studies that prove inconsistencies or lack of relationship between MHC diversity and characteristics related to evolutionary potential. There is also a number of studies proving that now, in the genomic era, studying MHC diversity alone is not enough to draw conclusions on species evolutionary potential.
Methods
- The authors performed multiple types of analyses that aimed to assess factors shaping DRB locus diversity, but some of them are not discussed in deep, and in many cases don’t add much to the understanding of processes shaping MHC diversity in studied populations. For example, they calculate nucleotide and amino acid distances between DRB sequences, and in the discussion, they only mention that obtained values are not different from the ones from French populations.
- Line 142; what type of Bayesian population structure analysis were performed? Are these results presented in the manuscript? I’m confused here.
- You state that there was no population structure between 10 populations or groups assigned according to geographical criteria. In this context, in my opinion, is pointless to analyse MHC diversity in these groups separately. Also, when you don’t see any differentiation at the population level, performing AMOVA test doesn’t add any more information.
- You present the relationships between roe deer DRB alleles using haplotype network, what is correct, but why didn’t you cluster DRB alleles into supertypes using positively selected sites? This could add valuable information on partitioning functional diversity in the region.
- Antigen binding sites were inferred from the human MHC II molecule structure. The results of the subsequent analysis that aimed to find positively selected sites in DRB locus don’t overlap well with these sites showing that ABS in roe deer are, at least to some extent, different. The approach that would be more informative here is to show which sites are pointed by both CodeML and individual codon tests. Those codons should be subsequently used for tests of historical selection and supertype clustering.
- Lines 198-203; did you consider detected recombination events while performing selection tests on the codons?
Results
- I’m a little concerned about the number of homozygous individuals, 47% is a high number. Homozygote excess was significant or close to significance in eight out of ten analysed populations. Homozygote excess is also visible while the data is analysed in division into three clusters, but the significance of HW test is not given. I’m afraid that this result may indicate non-amplifying DRB alleles that would affect the whole study.
- Line 259, line 279; the FST values are never negative, these values are actually zero.
- The differences in allelic richness between analysed groups are actually minor and figure 2 doesn`t add any valuable information.
Discussion
Lines 354-357; you state that your studied populations maintained a high level of MHC diversity and state that this is due to demographic fluctuations and founder events. This would rather lead to reduced diversity.
-line 372; I don’t agree that a lower number of pairwise differences among MHC alleles in studied populations is related to limited gene flow. MHC alleles normally present evolutionary old lineages, even with a high level of trans-species polymorphism and sequence diversity doesn’t reflect the current demographic process.
- lines 423-435; as I mentioned before, as there is no structure between 10 geographically assigned regions, the values describing its MHC diversity are not eligible. Secondly, the differences in diversity indices here are mostly minor and not statistically tested.
-lines 438; I don’t think there is a visible difference between the diversity of MHC region in the southwestern and northeastern part of the range (country) and other populations. Maybe this difference was more visible in microsatellite loci. I would rather argue that if this is so, MHC diversity is maintained in all populations despite demographic changes proved by microsatellites.
-lines 451-452; the most important outcome of this study is the difference between MHC and microsatellite structure and diversity what rather point to the action of balancing selection in maintaining MHC diversity. This is actually discussed earlier. I don’t understand why you write here that this is demography that shapes MHC diversity in your study.
-lines 464-480; this part should be significantly shortened.

Author Response
Dear Reviewer,
We are grateful for your attitude towards our manuscript and for very valuable comments and suggestions, but we do not agree that the manuscript is not of general interest to the scientific community. To our opinion, it indeed provides important information and knowledge on MHC allelic diversity in European roe deer in an area where such information is completely lacking.
This area represents the contact zone between two mitochondrial clades (Eastern and Central) of roe deer, and is/was particularly important for recolonization events in Europe (Randi et al. 2004, Plis et al. 2021). The limited distribution of the Eastern clade haplotype argues for the existence of a glacial refugium in the eastern areas, from which, however, roe deer did not spread far west. In contrast, the wide distribution of mtDNA haplotypes of the Central clade in the Balkan, Central and Northern Europe, the Apennines, the Alps and the Iberian Peninsula speaks for the existence of distinct refugial in southern areas. New and better insight into the pattern of genetic diversity of this widespread and important species is therefore also important to clarify the complex pattern of population structuring that likely results from historical vicariance in southern glacial refugia (as described mainly by mtDNA results) and subsequent mixing of populations due to natural secondary contacts or anthropogenic disturbance (as described mainly by STR data). For this reason, additional data – which are, in our study, also absolutely novel for the wider region of interest – on genetic diversity/outlook are relevant to solve complex pattern of roe deer recolonisation and admixture events in Europe.
We have addressed each comment individually; please find changes that we made and our responses point by point in attached file. In manuscript we highlight the changed parts.
We also made some additional minor editorial corrections in the manuscript.

Reviewer 2 Report
In this manuscript, Buzan et al. assess MHC diversity in roe deer in Slovenia. They made a valuable effort in analyzing MHC DRB exon 2 of Slovenian roe deer populations and discovered three new alleles. Authors found a lack of genetic structure and evidence of positive selection.
The authors present interesting immunogenetics results that contribute to the knowledge of relevant topics such as the relationship between host genetic diversity and the emergence and transmission of zoonotic diseases. It is a pity that this manuscript was not submitted to the published Animals’ special issue on “Genetic diversity of wild boar and deer”.
Despite the results of the manuscript can be publishable, there is a main drawback that might prevent its publication in the present form.
The comparison between the genetic structure at MHC loci and at neutral loci is a main objective of the work (L71-72). This issue is referenced in the simple summary and in the abstract. Moreover, it is covered in depth in Discussion (section 4.1). However, authors do not show any results regarding genetic structure at neutral markers. Contrarily, they used a published study (Buzan et al. 2020) as reference to interpret their results. Since the comparison is a main objective of this work, this comparison must be included as a result in the manuscript. Please, see L27 as an illustrative sentence for this main comment. I propose the following strategy:
The main authors of the manuscript are the same as those of the previously published paper (Buzan et al. 2020). I assume that authors can use both the MHC and the microsatellite genotypes. The samples used in Buzan et al. (2020) seem to be similar to those used in this manuscript. Buzan et al. (2020) used samples from 214 roe deer female (172 yearlings). I assume that those 172 yearlings are the same 172 female yearlings used in the present work. Therefore, authors would have 172 individuals genotyped with MHC and microsatellite markers. The dataset would not be the same as in Buzan et al. (2020) and analyses might also be slightly different. Authors could perform the same genetic structure analyses for both types of loci: Bayesian genetic population structure analysis (result in L142 must be included in a new version of the manuscript), genetic diversity measures, and genetic differentiation estimates. Consequently, they would present paired results similar to those currently shown in Table 2, Table S3, Figure 2, Table S4 and Table S5, for both MHC and neutral loci. The results of all these analyses would be included in sections 3.2. The results of the remaining analyses would remain in sections 3.1 and 3.2.
If this strategy is followed by authors, in addition to new results, authors must also include new information in Material and Methods section regarding microsatellite genotyping, but they may also cite Buzan et al. (2020) to reduce the length of the manuscript.
Author Response
Dear Reviewer,
We are grateful for your positive attitude towards our manuscript and for very valuable comments and suggestions. We have addressed each comment individually; please find changes that we made and our responses point by point in attached file.
We also made some additional minor editorial corrections in the manuscript.

Round 2
Reviewer 1 Report
I was asked to assess the corrected version of the manuscript "Neutral and selective processes shape MHC diversity in roe deer in Slovenia" by Elena Buzan, Sandra Potušek , Luka Duniš , Boštjan Pokorny. My first decision on the manuscript was reject, and unfortunately, I have to maintain this decision. In my opinion, the authors did not answer in full my concerns, and the changes they made do not raise the level of the manuscript. There are still major flaws that make this manuscript unpublishable in the present form. Below I present my comments on the most important issues using the answers given by the authors in the cover letter. - I appreciate the explanation of the phylogeographic context that gives an additional justification for describing MHC diversity in roe deer populations. Unfortunately, it was not used to place MHC description in the manuscript in the border context, nor Introduction neither Discussion. In the present form, the results are mainly descriptive and do not add much to species management or disease transmission knowledge. - Point 3. The sentences in lines 55-64 are very general and do not address specific issues related to studying only one MHC locus. These problems are presently well recognized in the literature. - Point 5. I still don't know what are Bayesian structural analysis. Do you mean software STRUCTURE? Also, finding no structure in the MHC is one of your important results. - Point 6. I have a great problem here. I went through the author's paper on microsatellite diversity performed virtually on the same individuals. First, microsatellite diversity shows clearly that the division into 10 groups is completely unsupported. Therefore you cannot present any genetic results for these groups (both microsatellites and MHC), because they do not exist. I'm afraid I also disagree with the second Reviewer's suggestion to rerun the microsatellite genotyping results. These results are already published and should not be analyzed for the second time performing the same type of analysis. It is also clearly seen in the results that are the same as obtained in the previously published paper. - Point 7. I agree that supertypes in MHC cannot always be determined. This is sometimes the case when a lower number of alleles is present. Nevertheless, this is important information that should be included in the manuscript. - Point 10. Here I meant that high and significant FIS values might be the problem of non-amplifying alleles that is technical problem at the laboratory stage. The primers you used are not specific for roe deer what may result in selective amplification of only a set of alleles. So if this is the case, using different approaches of validating genotyping results did not solve the problem. To be sure, you would need to redesign primers using roe deer intron sequences and regenotype your samples. Taking all these issues into account I cannot recommend this article for publication.Author Response
Please see the attachment

Reviewer 2 Report
The main concern I had about the previous version of the manuscript from Buzan et al. was that the work should present clear differences in genetic structure between MHC and neutral loci. This difference supports one of the main conclusions: “… MHC polymorphism was influenced primarily by balancing selection…” (L30; see also L553). That was why I proposed to include in the manuscript the results of neutral markers.
In this new version of the manuscript, Buzan et al. have included the analyses for nuclear markers. However, from my point of view, the presentation of results still needs to improve. There is a paragraph in Discussion that illustrate my current concern:
L454-467. “Clearly, MHC genetic diversity was primarily affected by balancing selection and to a lesser extent by neutral processes, with no evidence of local adaptation, which has already been suggested for roe deer by Quéméré et al. [25]. Our conclusion is supported by several facts that showed weaker patterns of genetic structuring for MHC compared to microsatellite loci.” Authors provide three facts that support this conclusion, but only one of them is clearly shown as a result of the manuscript (ii: pairwise Fst values provided by tables S4 and S7). Regarding (i), cluster analyses were not shown for MHC, and information for neutral markers is lacking (see below). Regarding (iii), a strong significant isolation by distance pattern was not shown for neutral loci (see below). Additionally, authors removed an interesting analysis (AMOVA) for which I expected to see a clear comparison between both types of markers.
Please, see specific comments for this concern:
L156-158. Why did not authors show the results of Bayesian genetic structure? There should be a graph in supplementary material that represent the values of deltaK as a function of the assessed Ks (see reference 71 (Evanno et al 2005)). If deltaK is maximized in a K value greater than 1, the membership coefficient of probability should be represented.
L179-181. Why did authors remove the AMOVA results? The comparison of AMOVA results between MHC and microsatellite loci would be evidence for balancing selection.
L273. Should the heading be changed considering that AMOVA analysis was removed? Anyway, I recommend maintaining the AMOVA result (see above).
L385. There should be a graph in supplementary material representing the values of deltaK as a function of the assessed Ks (see reference 71 (Evanno et al 2005)).
L386. The geographical distribution of clusters is not clearly represented with only the figure S3. Please, provide more information regarding the relationship between geographic origin and the membership coefficient of probability.
L446. In this new version of the manuscript, I expected to find that genetic differentiation was higher at microsatellite markers than in MHC DRB. This result might support the action of balancing selection over MHC loci. However, authors removed AMOVA analyses and now the result that support the balancing selection for MHC DRB is not clear. Pairwise Fst values between populations (Tables S4 and S7) might support the difference in the degree of genetic differentiation for both type of loci. Pairwise Fst values between clusters were also higher for neutral markers. However, these values were not significantly different from zero (as for MHC DRB). A comparison of AMOVA results for both MHC DRB and neutral markers might show additional evidence.
L465. Why did not authors assess isolation by distance with neutral markers? Figure S3 is not enough to interpret isolation by distance with microsatellite loci. Authors should conduct a Mantel test with neutral markers (as with MHC DRB).
Additional comments:
L13 and L25. Authors must modify these sentences because they conducted new analyses and the comparison was made with the results of this work. Perhaps, they might just remove “previously published”.
L468-473. Should this interpretation be based on allelic richness rather than number of alleles?
L484. I cannot see evidence of demographic changes in Table S5.
Table S5. Please, assess departures from HWE with microsatellite data.
L275. Please, actualize the reference 37.
Round 3
Reviewer 2 Report
In this new version of the manuscript, Buzan et al. followed the comments I provided for the previous version. From my point of view, the manuscript can be accepted for publication in Animals, after following some minor points.
Minos points:
1) STRUCTURE results for MHC genes (figure S4) should be mentioned in the Result section.
2) Authors should modify the sentence in L470 (“Clearly, MHC genetic diversity was primarily affected by balancing selection and to a lesser extent by neutral processes, with no evidence of local adaptation, which has already been suggested for roe deer by Quéméré et al.”). I am not completely convinced about this conclusion. Some results seem to support it (pairwise Fst values, isolation by distance…), but some doubts also remain. STRUCTURE and AMOVA results for MHC genes and microsatellite markers are not clearly different. Moreover, under a balancing selection for MHC genes, such a high difference between He and Ho (Table S3) would not be expected (departures from HWE were found for many populations). Note that for microsatellite markers Ho tend to be similar to He and no departure from HWE were found (Table S6). The first sentence in Conclusion section (L546-548) is much more appropriate.
Author Response
Dear Editor,
Thank you for your decision on minor revision. In accordance with this, we made additional corrections and improvements of the manuscript.
Best regards,
Elena Buzan
- In this new version of the manuscript, Buzan et al. followed the comments I provided for the previous version. From my point of view, the manuscript can be accepted for publication in Animals, after following some minor points.
Thank you for accepting the manuscript.
Minos points:
1) STRUCTURE results for MHC genes (figure S4) should be mentioned in the Result section.
We added this information into Results section (L. 291-292).
2) Authors should modify the sentence in L470 (“Clearly, MHC genetic diversity was primarily affected by balancing selection and to a lesser extent by neutral processes, with no evidence of local adaptation, which has already been suggested for roe deer by Quéméré et al.”). I am not completely convinced about this conclusion. Some results seem to support it (pairwise Fst values, isolation by distance…), but some doubts also remain. STRUCTURE and AMOVA results for MHC genes and microsatellite markers are not clearly different. Moreover, under a balancing selection for MHC genes, such a high difference between He and Ho (Table S3) would not be expected (departures from HWE were found for many populations). Note that for microsatellite markers Ho tend to be similar to He and no departure from HWE were found (Table S6). The first sentence in Conclusion section (L546-548) is much more appropriate.
We rephrased the sentence accordingly (L. 273-275).
